# Multi-Ring Disk Resonator with Elliptic Spokes for Frequency-Modulated Gyroscope

**DOI:** 10.3390/s23062937

**Published:** 2023-03-08

**Authors:** Shihe Wang, Jianlin Chen, Takashiro Tsukamoto, Shuji Tanaka

**Affiliations:** 1Department of Robotics, Tohoku University, Sendai 9808579, Japan; wang.shihe.t2@dc.tohoku.ac.jp (S.W.); mems@tohoku.ac.jp (S.T.); 2School of Microelectronics, Shanghai University, Shanghai 201800, China; 3Micro System Integration Center (µSIC), Tohoku University, Sendai 9808579, Japan

**Keywords:** disk resonator, mode matched resonator, frequency mismatch, (100) single crystal silicon, multi-ring structure, elliptic spoke

## Abstract

In this paper, we report a multi-ring disk resonator with elliptic spokes for compensating the aniso-elasticity of (100) single crystal silicon. The structural coupling between each ring segments can be controlled by replacing the straight beam spokes with the elliptic spokes. The degeneration of two *n* = 2 wineglass modes could be realized by optimizing the design parameters of the elliptic spokes. The mode-matched resonator could be obtained when the design parameter, aspect ratio of the elliptic spokes was 25/27. The proposed principle was demonstrated by both numerical simulation and experiment. A frequency mismatch as small as 1330 ± 900 ppm could be experimentally demonstrated, which was much smaller than that of the conventional disk resonator, which achieved as high as 30,000 ppm.

## 1. Introduction

Micro-Electro-Mechanical Systems (MEMS) gyroscopes have widespread applications in many fields; they could be used for stabilization and rollover detection in smartphones and drones, or be applied together with accelerometers for inertial navigation purposes, and also have a wide range of application in robots and military applications, etc. They have become increasingly popular over the last 20 years due to their advantages, such as small volumes, stable performance, and relatively low costs [1,2,3]. In MEMS gyroscopes, a resonator is applied for sensing the Coriolis force. Based on the utilization of the signals from the resonators, the Coriolis vibratory gyroscopes can be classified into several types. Amplitude-modulated (AM) and force-to-rebalanced (FTR) gyroscopes have been studied for a long time. The AM gyroscopes detect the Coriolis force through the amplitude signal of the resonator, but this method imposes a trade-off between sensitivity and operation bandwidth [3,4]. FTR gyroscopes are proposed to solve this problem, which retain the amplitude of the sensing axis static through the feedback control, possess an increased bandwidth, and improve the detection stability [5]. However, the scale factor is sensitive to the temperature, and the size of the supply voltage also limits the range of this control method [6].

To overcome the problems in those gyroscopes, frequency-modulated (FM) gyroscopes have been proposed to improve the performance [7,8], which detect the Coriolis force as the frequency change. For over one decade, FM gyroscopes have been in development and were seen as a better choice for commercial gyroscopes in the future [7]. In FM gyroscopes, removing the common-mode disturbance, which mainly comes from the temperature fluctuation, two resonant modes are used, the frequency of which is modulated by the Coriolis force in opposite directions. In real applications, FM gyroscopes convert the external angular rate into a change in the resonance frequency, and the gyroscope self-oscillation loop contains information on the input angular rate. Therefore, the frequency read-out module of FM gyroscopes is critical [9]. Until now, there have been a variety of technologies proposed for FM detection, such as using two resonators [10], mode-splitting [11], time-switching [12], and Lissajous-FM (LFM) control [9,13,14]. With the exception of the LFM versions, which work under the condition that mode split exists, all FM gyroscopes require a degenerated resonator, in which two axes have the same resonant frequency and quality factor (Q-factor) [15,16]. Until now, lots of structures have been proposed for the degenerated resonator, such as quadruple mass [17,18], dual mass [19,20,21], single ring [22,23], hemispherical or dual shell [24,25,26], and multi-ring (disk) [27,28] resonators.

Ring and multi-ring (disk) resonators are advantageous in terms of their structural simplicity and low immunity for acceleration due to the symmetrical structure. Compared with the hemispheric gyroscope proposed in recent years, which has the best performance among vibratory gyroscopes, ring or disk resonators have unique advantages in terms of the fabrication difficulty for their two-dimensional structures [29,30]. Between ring and disk types, the disk resonator has a better performance than the ring resonator because the disk structure characteristics possess higher detection sensitivity, Q-factor, and better electrostatic tuning ability [31,32].

However, two *n* = 2 wineglass modes of the ring-type resonator, as shown in Figure 1, made of (100) single crystal silicon (SCS), have different resonance frequencies due to the anisotropic stiffness. To overcome this problem, isotropic materials such as polycrystalline silicon [33,34] or (111) SCS [35,36] have been used. However, the (100) SCS is widely used for MEMS and is advantageous in terms of its process compatibility and integration with the other devices. As stated above, ring or disk resonators have unique advantages in their fabrication process and bringing new materials into the fabricate will cause them to lose such advantages over other vibratory gyroscopes. Therefore, to avoid bringing new materials into the fabrication, some techniques have attempted to solve the problem of the structure design of the resonator, such as introducing the angle dependent perturbation into the resonator parameters [37,38], but these methods introduce the asymmetry into the design and bring a mismatch between the two modes, and for application in real gyroscope products, the robustness and stability of design are very important.

Therefore, in this paper, a new design of a degenerated disk resonator is proposed for the FM gyroscopes, which overcomes the material anisotropy problem of (100) SCS, without bringing asymmetry into the design. This proposed design has been confirmed by numerical studies and a series of experiments, including a comparison group.

## 2. Materials and Methods

This section provides a concise statement of the proposed design and working principle, together with finite element method (FEM) simulation results.

### 2.1. Working Principle

The two *n* = 2 wineglass modes, the <100> (Mode 1) and <110> (Mode 2) directions, as shown in Figure 1, have different resonant frequencies due to the stiffness anisotropy of the (100) SCS. Mode 2 of the single-ring resonator has a higher frequency than that of Mode 1, as shown in Figure 2a [37,39]. On the other hand, the resonant frequencies of Mode 1 and Mode 2 in a multi-ring (disk) resonator, in which straight interconnection beams are used to couple each ring, show the opposite order [38,40], as shown in Figure 2b. As shown in Figure 1, the deformation of each ring is distorted by the connection spokes. This mechanical constraint increases the effective stiffness of the whole resonator structure, and this mechanical hardening effect affects Mode 1 rather than Mode 2, thus the frequency order switches. In other words, the mode-mismatch of the single ring resonator, which has a higher resonance frequency in Mode 2 than that of Mode 1, was overcompensated by too much mechanical coupling between the rings.

From these phenomena, a novel structure to eliminate the stiffness anisotropy is proposed. The basic idea is that the anisotropy of each ring is compensated by weak coupling interconnections. In this study, elliptic shape spokes, as shown in Figure 3, are used to couple each ring, instead of the linear spokes used in the previous disk resonators. An elliptic shape is a steady structure and has smooth edges, which could reduce the effect from the fabrication error; other bending structures, such as a tetragonal ring and crank [41,42] shapes, would also be possible effective choices for future study.

The stiffness of the spoke could be controlled by the width, *v*, and aspect ratio, *b*/*a*. The longitudinal compression stiffness of the conventional linear beam can be express as:
(1)$$k_{b}=\frac{Evt}{l}$$
where *E*, *v*, *t*, and *l* are the (100) SCS’ Young’s modulus value, the width of linear beam (as shown in Figure 3), the thickness of device, and the length of linear beam, respectively. Therefore, for the conventional disk resonator, the only way to decrease the linear beam’s stiffness is to reduce the width and height of it, but this method has limitations according to the design rule. In our simulation result, even the width of the linear beam was reduced to 5 µm and the mismatch of frequency was still as large, at around 41,800 ppm, demonstrating that the improvement in the parameter change is very small for a linear beam structure.

On the other hand, for the elliptic spoke, the compression stiffness of each part can be roughly approximated as follows. The ellipse is divided into upper and lower parts, and each part was approximated as the linear beam. When assuming that both sides of the beam are simply supported and the force is applied to the center of beam, the stiffness is given by:(2)$$k_{c}=\frac{48EI}{L^{3}}$$
where *I* and *L* (= 2*b*) are the geometrical moment of inertia and length of the beam, respectively. The geometrical moment of inertia, *I*, is expressed as:(3)$$I=\frac{t{\left(\frac{v}{2}\right)}^{3}}{12}=\frac{tv^{3}}{96}$$

Thus, the total effective stiffness could be expressed as:(4)$$k_{e}=2\times \frac{48E\frac{tv^{3}}{96}}{{\left(2b\right)}^{3}}=\frac{Etv^{3}}{8b^{3}}$$

Compared with the conventional linear beam (Equation (1)), the stiffness of the elliptic spoke could be effectively adjusted by the design parameter, *b*. This means the mechanical coupling between rings could be effectively adjusted by the elliptic spokes. The discussion here is based on approximation, but it could explain the qualitative principle. Precise estimation is conducted by numerical simulation, as follows.

### 2.2. Numerical Simulation

To confirm the precise effect of the elliptic spoke, the resonant frequencies are simulated using the finite element method (FEM).

#### 2.2.1. Resonant Frequencies

To confirm the principle, two multi-ring resonators, one of which has the conventional linear spokes and the other has the elliptic ones, were designed and compared. The (100) SCS was used as the structural material, and the <100> direction of the SCS is set along the *X*-axis. The resonators consist of one central electrode and ten concentric rings. Eight interconnection spokes are designed symmetrically. The width of the rings was designed at 10 µm. The width and length of the linear spoke were designed at 10 µm and 50 µm, respectively. The width of the elliptic spoke was designed at 5 µm, and the aspect ratio, *b*/*a*, was selected as the parameter.

The modal shapes are shown in Figure 4. The deformation of the ring in the conventional resonator, especially the outer ring, was highly distorted by the rigid linear spokes. On the other hand, the resonator with elliptic spokes showed less distortion due to the moderate coupling stiffness given by the elliptic spokes.

The design parameters and simulation results are summarized in Table 1. The resonant frequencies of the conventional resonator (Figure 4a) are 82,090 Hz (Mode 1) and 78,545 Hz (Mode 2), which means the frequency mismatch was 44,000 ppm. Figure 5 shows the dependency of the resonant frequencies on the design parameter *b* (parameter *a* was fixed at 27 µm, the distance between rings was always 50 µm, as shown in Table 1, and the total length was 54 µm for the elliptic spoke to connect with the rings). The resonant frequencies of both modes gradually decrease with the increase in the aspect ratio. However, the frequency of Mode 1 changes more rapidly than that of Mode 2, which means that the aspect ratio could be used for tuning the frequency mismatch. In the FEM results, the optimized aspect ratio, *b/a*, was found at 25/27, where the calculated frequency mismatch at this aspect ratio was estimated to be as small as 70 ppm.

#### 2.2.2. Quality Factors

Usually, the Q-factor of the disk resonator is limited by thermoelastic dissipation (TED). The TED-related Q-factor, Q_TED_, of the elliptic spoke resonator was estimated by FEM [43]. Figure 6 shows the estimated Q-factors with the different design parameters, *b*.

The Q_TED_ increases gradually with a larger aspect ratio. The thermal time constant of a 10 µm wide ring is about 110 ns (9 MHz), and thus the resonator is operated under the isothermal region. Therefore, the frequency reduction caused by design parameter *b* (a is fixed) increases the Q_TED_. For that in the isothermal region, 1/Q_TED_ is proportional to the frequency [44]. The Q-factor mismatch at an aspect ratio of 25/27 was 1.7%. The Q-factor mismatch of the resonator with linear beams was estimated as 13%, which was also generated by the frequency mismatch. From these results, the proposed method not only minimizes the frequency mismatch, but also the Q-factor mismatch, thus it is suitable for the FM gyroscopes.

#### 2.2.3. Robustness against Fabrication Error

There are two major error sources that result from the manufacture process: the orientation error in the SCS and the width change of the concentric rings during fabrication. The effect of them on the frequency mismatch was studied. The optimized design, i.e., an aspect ratio of 25/27, was used as the resonator design. A mismatch parameter, *θ*_mis_, which was defined as the angle difference between the Mode 1 principal axis and the <100> direction of the SCS, was used to simulate the angle error. The frequency mismatch is defined as:(5)$$\Delta f=\left|\frac{f_{Mode1}-f_{Mode2}}{f_{Mode1}+f_{Mode2}}\times 2\times 10^{6}\right|$$
where *f* represents the resonant frequency of Mode 1 or 2. Figure 7 shows the calculated dependency of the frequency mismatch on the angle error, *θ*_mis_. In the proposed design, the angle error of the SCS has a relatively small effect on the frequency mismatch, which might come from the symmetric structure. This means that the designed resonator with elliptic spokes has good robustness towards the angle error in the SCS material.

Figure 8 shows the obtained dependency of the frequency mismatch on the errors of the ring and spoke widths. In this analysis, the angle error was neglected, i.e., *θ*_mis_ = 0° is assumed. Frequency matching could be obtained when the ring width was 10 µm and the spoke width was 5 µm. Both the mode frequencies decrease or increase simultaneously when the ring and spoke widths become narrower or wider due to the change in the effective stiffness, as the symmetric graph shows in Figure 8. However, Mode 1 exceeds Mode 2 when the ring width is less than 10 µm. On the contrary, Mode 2 becomes larger when the spoke width is less than 5 µm. Therefore, if the widths of the ring and spoke change at the same rate, i.e., both stiffnesses decrease or increase, the frequency mismatch is less than 1000 ppm even when the structural error exists, as shown by the dark part in Figure 8.

From these results, it is clear that the proposed method is strong against fabrication errors.

## 3. Results

To demonstrate the effectiveness of the proposed method, two types of resonators were fabricated and evaluated, one of which has the elliptic spokes and the other has the linear spokes. The fabrication process was based on the standard silicon on insulator (SOI) process, as shown in Figure 9.

### 3.1. Device Fabrication

The resonators are fabricated from a SOI wafer with a 50 µm Si device layer, 2 µm SiO_2_ box layer, and 450 µm Si handle layer. Firstly, a 20/300 nm thickness of the Cr/Au films are deposited on the device layer using RF magnetron sputtering. In contrast to the previous fabrication process [45], a new developed metal deposition method was applied in the fabrication process for the optimized design [46]. In the previous fabrication, the quality of the electrodes always limits the performance of the resonator, and this problem has been solved through the application of our new developed metal pads fabrication technology, as shown in Figure 10.

After the metal deposition, a 3 µm thick photoresist (OFPR800LB) was deposited and patterned. Then, the metal films were patterned by wet etching. After removing the photoresist, another layer of 3 µm thick photoresist (OFPR800LB) was deposited and patterned to define the resonator. The device layer was then etched through by deep reactive ion etching (DRIE). Figure 11 shows the device after the DRIE. To prevent over-etching due to the micro-loading effect, padding structures were placed in between the rings, as well as inside the elliptic spokes. Then, each resonator was separated by blade dicing. Finally, the resonator part, supported by the central electrode, was released via vapor phase HF etching (VPE). Figure 12 shows the scanning electron micrographs (SEMs) of the resonator.

### 3.2. Evaluation

In this study, to demonstrate the stability and improvement of the proposed design, several devices with optimized design are evaluated, and the resonator with a linear beam is also evaluated as the comparison group. The experimental setup used in this study is shown in Figure 13. An actuation signal was applied on the surrounding electrodes to actuate the resonators, and the oscillation was detected by a Laser Doppler Vibrometer (LDV). The detected signal was demodulated by the lock-in amplifier. These experiments were conducted under a vacuum condition less than 5 Pa.

Figure 14 shows the frequency responses of the conventional resonator. Figure 14a,b show the measurement results of the <100> (Mode 1) and <110> (Mode 2) modes, respectively. From the figures, it is clear that the observed modes were *n* = 2 wineglass modes. The resonant frequencies of Modes 1 and 2 were 67.086 kHz and 65.104 kHz, respectively, which means the frequency mismatch was 30,000 ppm.

Figure 15 shows the measurement results of the resonators with the elliptic spokes. It is clear that the *n* = 2 wineglass modes were successfully obtained. The resonance frequencies of Modes 1 and 2 were 66.218 kHz and 66.097 kHz, respectively, which shows a frequency mismatch of 1830 ppm.

To confirm the variety of the fabrication error, three resonators were evaluated. The results are summarized in Table 2. The observed frequency mismatches were 1830 ppm, 580 ppm, and 1570 ppm. The detected resonant frequencies data of Devices 2 and 3 are shown in Figure 16. The average value was 1326 ppm, and the 90% confidential interval was 900 ppm. The observed values were higher than the design value estimated by the FEM (70 ppm). This difference might be the result of the fabrication error, as shown in Figure 8. On the other hand, the frequency mismatch of the conventional resonator was measured as 30,000 ppm, which was a little bit smaller than the FEM prediction (44,000 ppm).

## 4. Discussion

This study aimed to solve the anisotropy problem caused by (100) single crystal silicon, a material widely used in mass product fabrication, and proposed an effective design for high performance FM gyroscopes. Our results revealed a frequency mismatch as small as 1330 ± 900 ppm by a series of experiments. As a comparison, a high frequency mismatch of 30,000 ppm was evaluated using resonators with a conventional structure.

Considering other recent research works that have explored improving the design for disk resonators, the results obtained in this study show many advantages in the working performance. Y. Shu et al. [37] reported a geometrical compensation method for a (100) silicon ring resonator. Although they obtained an improvement in terms of decreasing the frequency mismatch between two resonant modes, the best result under electrical tuning was still 54 Hz, which is even larger than the result obtained in Device 2 (31 Hz) in this study, which was evaluated without any tuning method. If the eigenfrequencies of two modes are considered, the performance difference between two designs will be much larger. In addition, a good robustness against fabrication error and the stability of the devices are demonstrated in this study.

With the exception of the frequency mismatch, the observed Q-factor mismatch was evaluated in real experiments, which was larger than the simulation results of Q_TED_, and no obvious difference was found between the conventional and the proposed resonators. However, the total Q-factor was affected not only by the TED, but also by other factors [43], such as anchor-loss, squeeze file damping, etc. Thus, this difference might be a result of the structural asymmetricity.

In the future, the small remaining frequency mismatch, which is less than 2000 ppm, and the Q-factor mismatch could be compensated by electrostatic tuning [16,47] or feedback control [11,48]. Considering the present remaining frequency mismatch and the limitation range of the tuning method, a reduced frequency mismatch as small as 0 could be reported [38], which can completely meet the requirement for FM gyroscopes.

Despite some improvement space for future work, this study successfully demonstrated that the proposed method could effectively compensate the (100) SCS anisotropic stiffness in the multi-ring disk resonator. The series of experiments show the good stability and practicability of the proposed design, which are required for the real application of FM gyroscopes or rate integrating gyroscopes [27,49,50], which also require the degenerated resonators.

## 5. Conclusions

In this paper, a novel multi-ring disk resonator for the high-performance FM gyroscope is proposed, in which the rings were weakly connected by elliptic spokes. The working principle was confirmed by both the numerical method and experiments. The shape of the elliptic spoke modifies the mechanical coupling between the rings, and thus the stiffness anisotropy can be compensated. The shape of the elliptic spoke was optimized by the FEM. A frequency mismatch as small as 1330 ± 900 ppm was obtained for the proposed device, which was much smaller than that of the conventional resonator, which consists of multiple rings and linear spokes, as high as 30,000 ppm. These results indicate that the stiffness anisotropy could be successfully compensated by the proposed method. The remaining frequency and Q-factor mismatch could be compensated through electrostatic tuning or feedback control. Thus, the proposed method could be applicable for FM gyroscopes, as well as rate integrating gyroscopes.

## Figures and Tables

**Figure 1 sensors-23-02937-f001:**
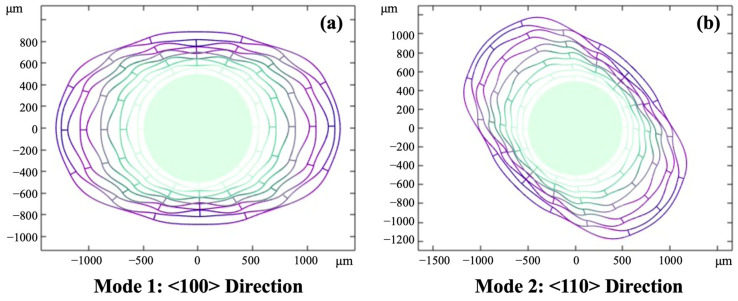
Wineglass vibration modes of a multi-ring disk resonator along (**a**) <100> direction (Mode 1) and (**b**) <110> direction (Mode 2).

**Figure 2 sensors-23-02937-f002:**
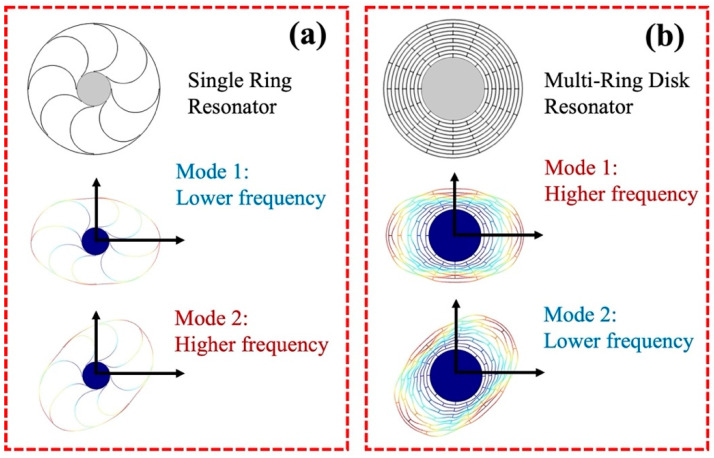
Frequency order between Mode 1 and Mode 2, in (**a**) single-ring resonator and (**b**) multi-ring disk resonator.

**Figure 3 sensors-23-02937-f003:**
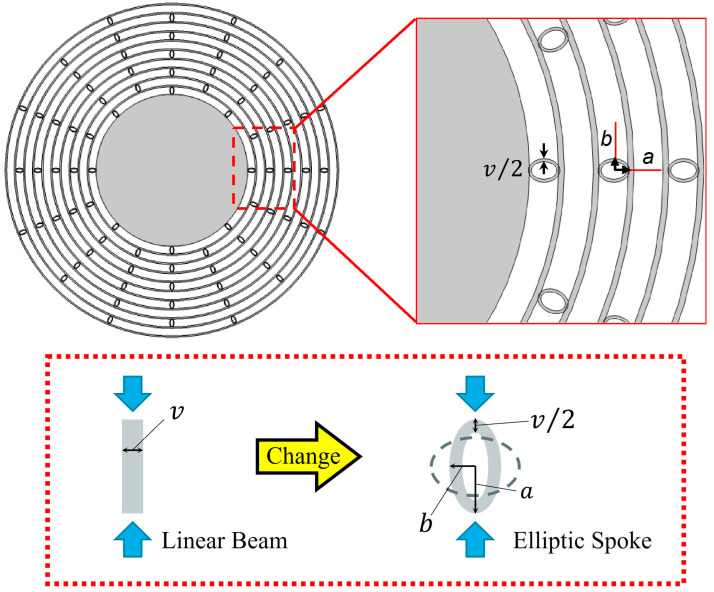
Schematic and some key parameters of elliptical spoke resonator.

**Figure 4 sensors-23-02937-f004:**
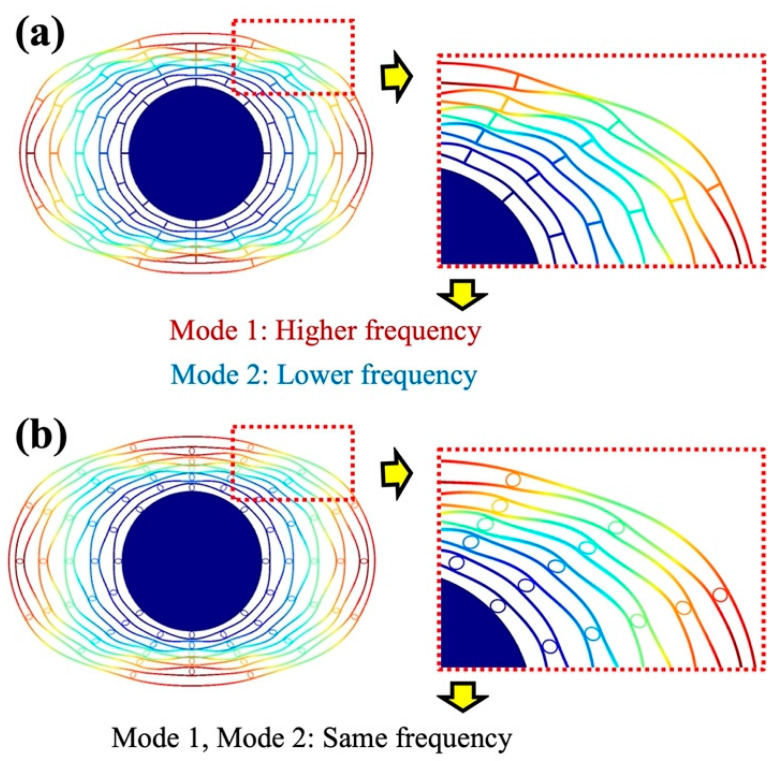
<100> mode shapes deformation in FEM simulation, the distortions of 45° segments are weakened in (**b**) proposed resonator with elliptic spokes, compared with (**a**) conventional resonator with linear beams.

**Figure 5 sensors-23-02937-f005:**
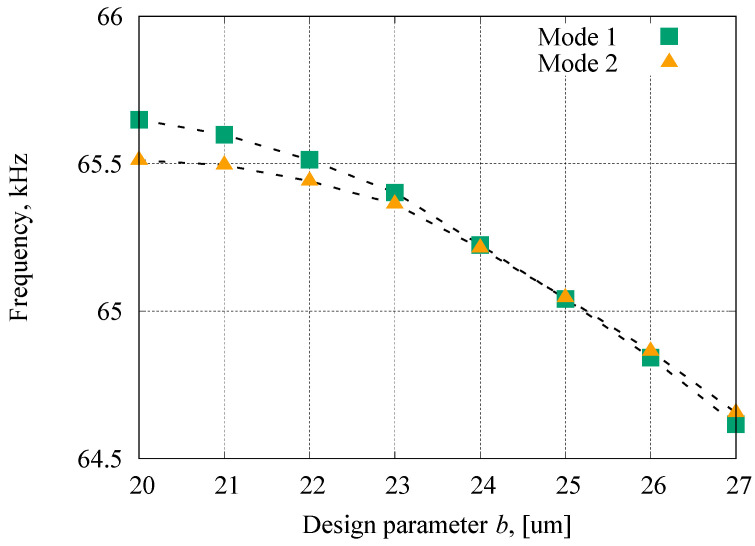
Dependency of resonance frequencies of Mode 1 (<100> direction) and Mode 2 (<110> direction) on the design parameter *b*.

**Figure 6 sensors-23-02937-f006:**
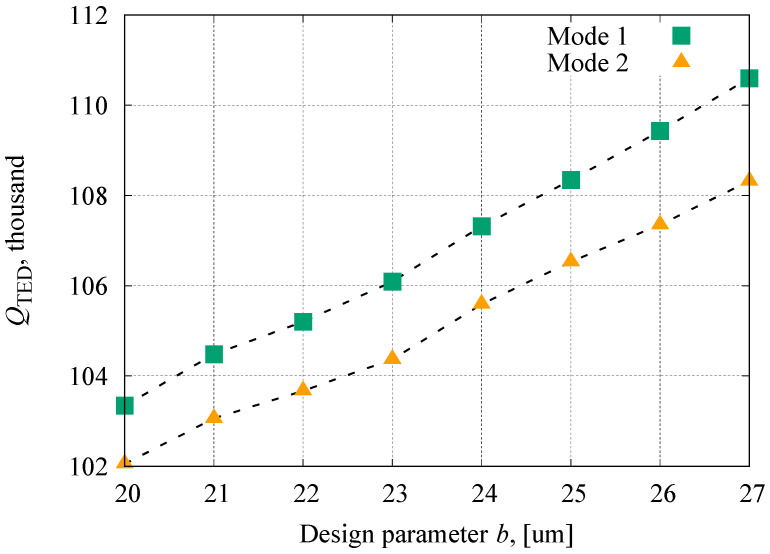
Q_TED_ of Mode 1 and Mode 2 under different parameter *b*.

**Figure 7 sensors-23-02937-f007:**
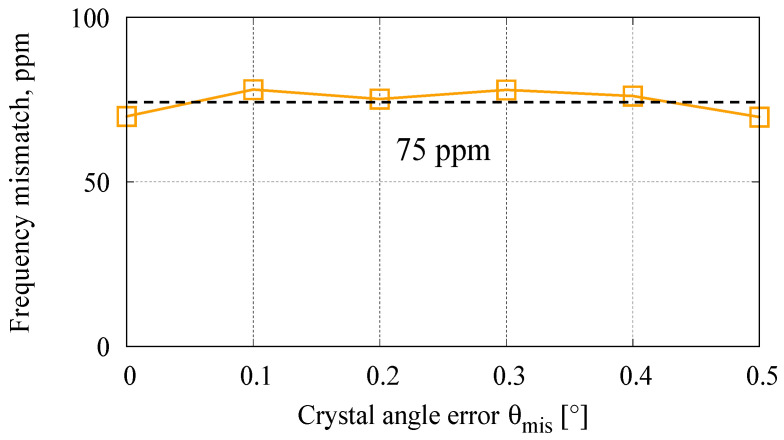
Frequency mismatch between two resonant modes (in the optimized design) under different angle error of crystal Si.

**Figure 8 sensors-23-02937-f008:**
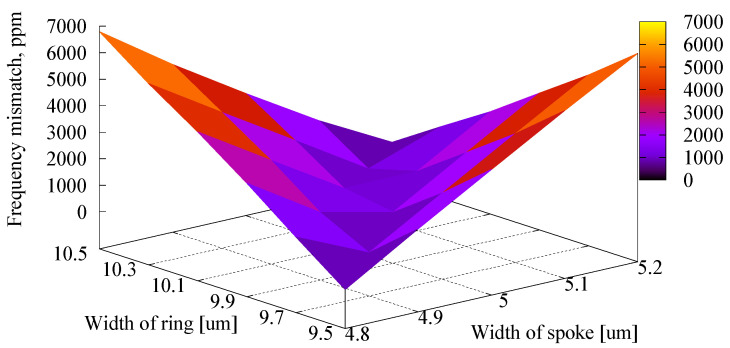
Frequency mismatch between two resonant modes (in the optimized design) under different ring width and spoke width error.

**Figure 9 sensors-23-02937-f009:**
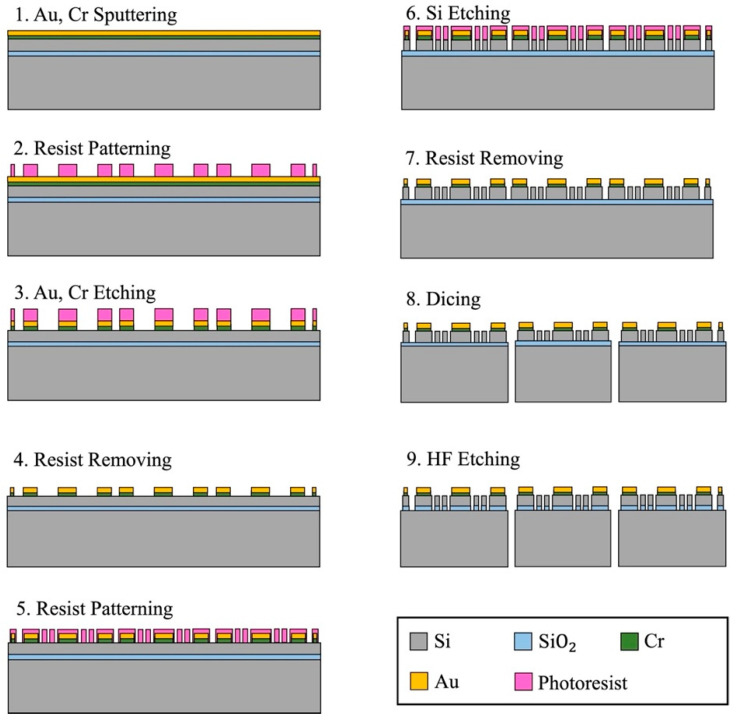
Schematic flow of MEMS manufacture process for the resonators, which was developed based on the standard SOI process.

**Figure 10 sensors-23-02937-f010:**
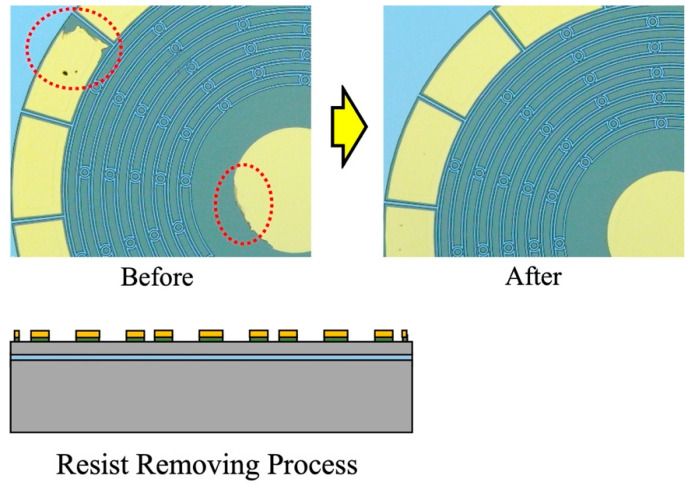
Improvement on fabricated electrodes after applying developed deposition technique.

**Figure 11 sensors-23-02937-f011:**
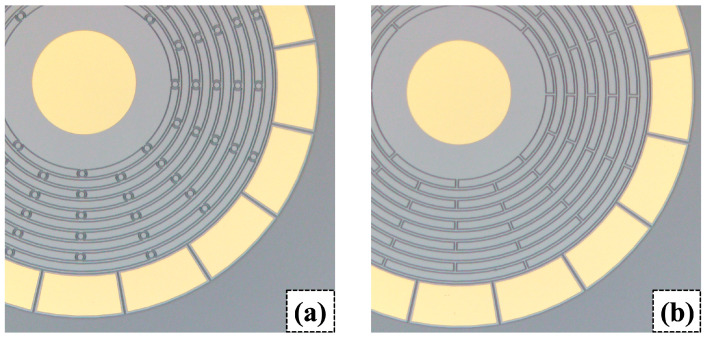
Manufactured resonant structure and electrode pads of (**a**) designed resonator with elliptic spokes, and (**b**) conventional resonator with linear spokes under microscope.

**Figure 12 sensors-23-02937-f012:**
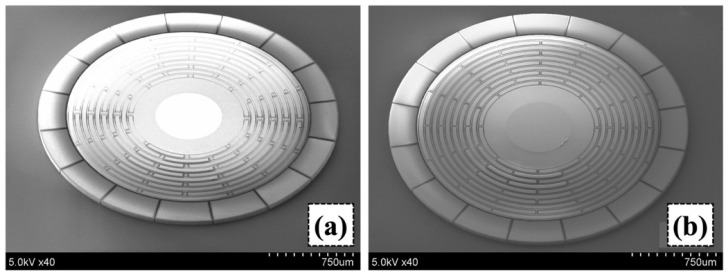
SEM images showing three-dimensional manufactured (**a**) designed resonator with elliptic spokes, and (**b**) conventional resonator with linear spokes.

**Figure 13 sensors-23-02937-f013:**
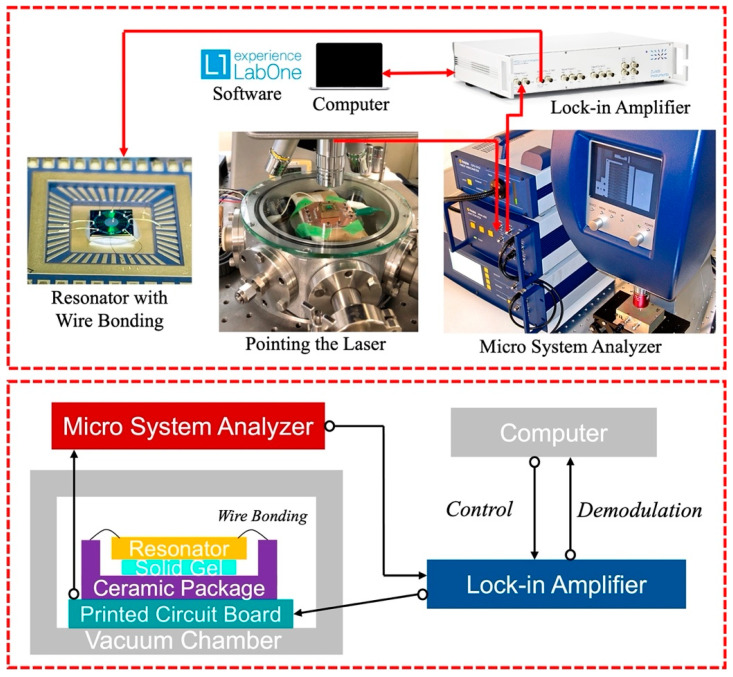
Schematics of applied evaluation system.

**Figure 14 sensors-23-02937-f014:**
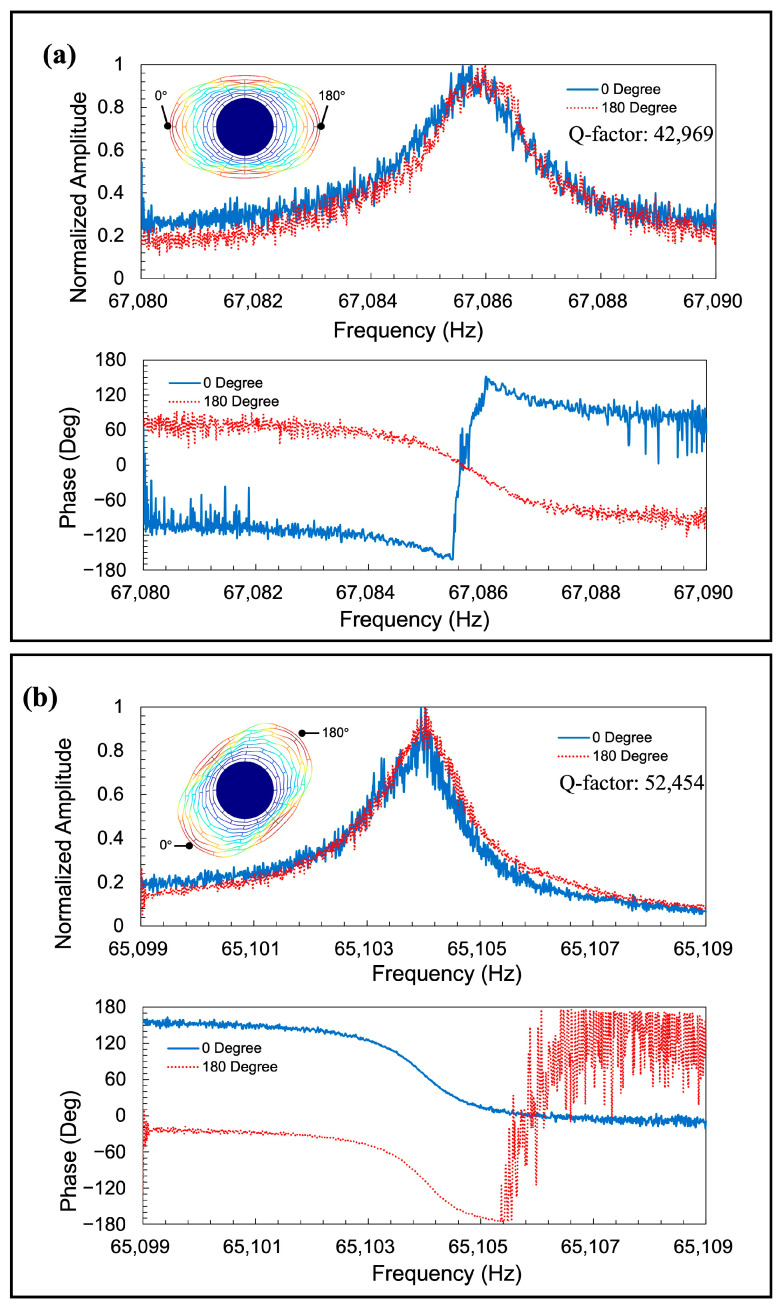
Detected amplitude change and phase change from actuated resonator with linear beams in (**a**) Mode 1 and (**b**) Mode 2.

**Figure 15 sensors-23-02937-f015:**
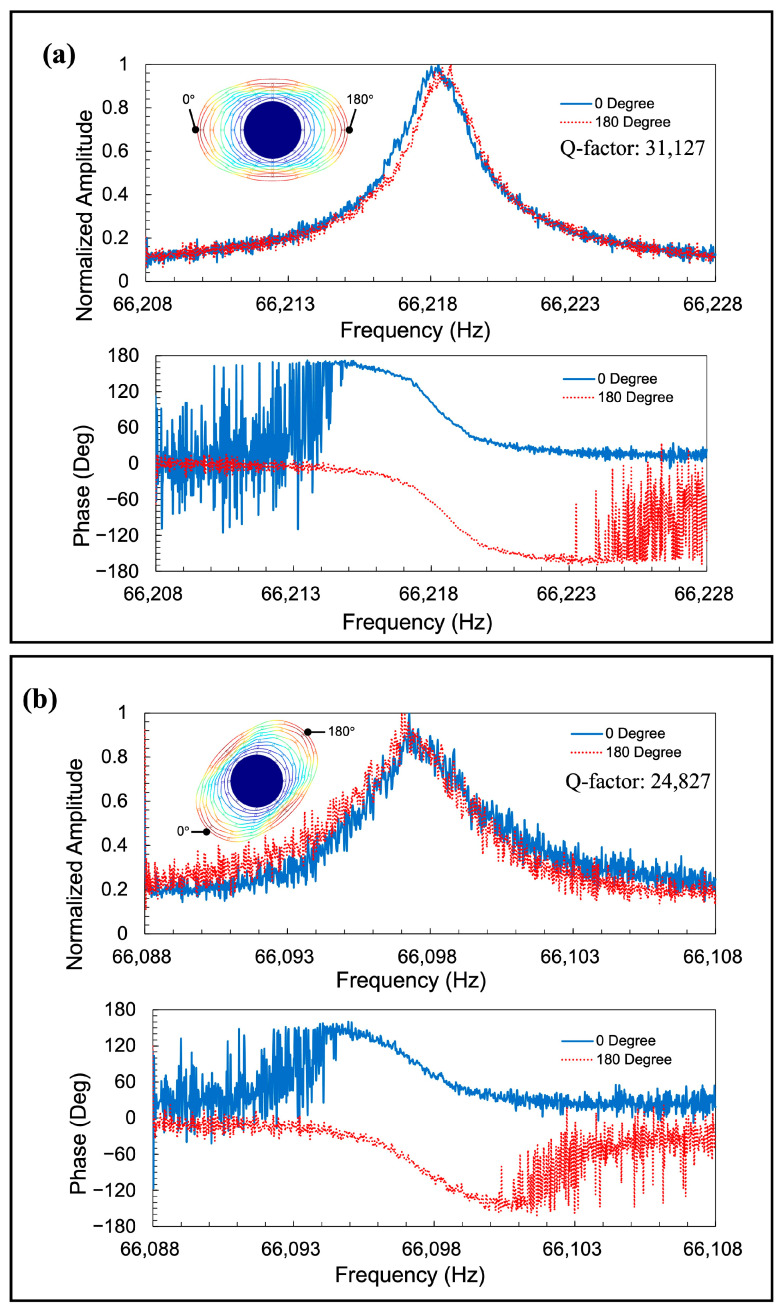
Detected amplitude change and phase change from one actuated resonator with elliptic spokes in (**a**) Mode 1 and (**b**) Mode 2.

**Figure 16 sensors-23-02937-f016:**
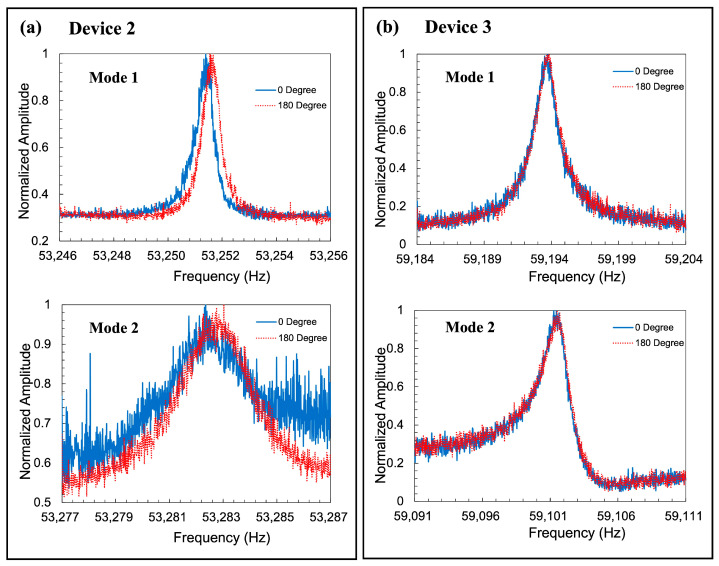
Detected amplitude change in Mode 1 and Mode 2 of (**a**) resonator 2 and (**b**) resonator 3 with elliptic spokes. All of the devices have the same design parameters.

**Table 1 sensors-23-02937-t001:** Design parameters and simulation results.

	Proposed Resonator (with Elliptic Spokes)	Conventional Resonator (with Linear Beams)
Thickness	50 µm	
Density	2330 kg/m^3^	
Thermal Conductivity	130 W/m/K	
Number of Rings	10	
Width of Rings	10 µm	
Distance between Rings	50 µm	
Radius of Centre Structure	500 µm	
Resonant Frequency Mismatch	70 ppm	44,000 ppm

**Table 2 sensors-23-02937-t002:** Experimental results of fabricated conventional resonator and proposed resonator.

	Proposed Resonator (with Elliptic Spokes)			Conventional Resonator (with Linear Beams)
Device thickness	50 µm			
Width of rings	10 µm			
Device ID	1	2	3	1
Q-factor mismatch	25.4%	26.2%	30.4%	22.1%
Mode 1 frequency	66.218 kHz	53.252 kHz	59.194 kHz	67.086 kHz
Mode 2 frequency	66.097 kHz	53.283 kHz	59.101 kHz	65.104 kHz
Frequency Mismatch	1830 ppm	580 ppm	1570 ppm	30,000 ppm

## Data Availability

Not applicable.

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
