# Peer review of "Multi-Ring Disk Resonator with Elliptic Spokes for Frequency-Modulated Gyroscope"

_sensors, 2023, doi:10.3390/s23062937_

Round 1
Reviewer 1 Report
Very interesting article, thank you very much. Do not see anything to be improved or corrected.
1. What is the main question addressed by the research?
Authors propose a novel idea of elliptic shape for spokes that connect rings in a multi-ring disk resonator. This solution helps to overcome the problem of frequency mismatch, caused by anisotropy of the whole structure, inherited from the anisotropy of silicon itself. All in all it helps to improve the accuracy of micromechanical gyroscopes. 2. Do you consider the topic original or relevant in the field? Does it
address a specific gap in the field?
Yes, it definitely is. In the results section authors clearly present the positive effect of their idea implementation, and I've never seen any other effective solutions for the stated problem. 3. What does it add to the subject area compared with other published
material?
As far as I know, it is the first time that this kind of spokes is proposed and this solution is the most effective one. Also it is shown that the method is not only theoretically effective, but is ready to be introduced to fabrication processes. 4. What specific improvements should the authors consider regarding the
methodology? What further controls should be considered? Do not see any problems regarding the methodology.
5. Are the conclusions consistent with the evidence and arguments presented
and do they address the main question posed?
Yes, they are. 6. Are the references appropriate? Yes, they are.
7. Please include any additional comments on the tables and figures. Everything is fine with figures and tables.
Author Response
We would like to thank you for facilitating the timely and high quality review process. Your comments and suggestions are very valuable to improve the quality of our manuscript.
Reviewer 2 Report
The paper is interesting and well written, however there are some points which deserve some clarification/revision.
1. The Authors propose to link the rings of a multi-ring resonator by elliptical spokes instead of straight ones to reduce frequency mismatch. Of course this new solution gives a less stiff connection, as it relies on bending instead of axial stiffness. However the Authors do not motivate the choice of the elliptic shape: why this shape is convenient with respect to any other connection working in bending ?
2. The stiffness of the straight spokes can be modified by changing the length and the thickness. The comparison between the new and old configuration is a bit unfair as in the new configuration the thickness is one half (and hence probably more subject to imperfections) and the length is higher (54 microns instead of 50). Please add a comment
3. The analytical evaluation of the equivalent stiffness, based on straight beams is very rough. Actually it is possible to properly evaluate the stiffness using beam theory of curved beams.
4 Figure 8 is not very clear from a graphical point of view: the point of frequency matching cannot be detected.
5 There are a lot (10 out of 39!!) self-citations. Some of them are only loosely related to the contents of the paper. I strongly suggest to reduce the number of self-citations.
Author Response
- The Authors propose to link the rings of a multi-ring resonator by elliptical spokes instead of straight ones to reduce frequency mismatch. Of course this new solution gives a less stiff connection, as it relies on bending instead of axial stiffness. However the Authors do not motivate the choice of the elliptic shape: why this shape is convenient with respect to any other connection working in bending ?
Thank you for your nice comments. According to your comments, we would like to add other choice of bending beams and more FEA results in the manuscript: (page 3 line 93-95 and page 4 line 106 to 108)
Elliptic shape is steady structure and has smooth edges which could reduce the effect from fabrication error, of course other bending structures such as tetragonal ring and crank [29, 30] shapes would also be possible effective choice for future study.
In our simulation result, even the width of linear beam was reduced to 5 µm, mismatch of frequency was still as large as around 41800 ppm, demonstrating that the improvement with parameter change is very small for linear beam structure.
- The stiffness of the straight spokes can be modified by changing the length and the thickness. The comparison between the new and old configuration is a bit unfair as in the new configuration the thickness is one half (and hence probably more subject to imperfections) and the length is higher (54 microns instead of 50). Please add a comment
According to your comments, we did another simulation using the straight spokes with 5 µm, which show the frequency mismatch was as large as 41800 ppm. The simulated result showed the elliptic spoke can effectively compensate the intrinsic aniso-elasticity. We would like to add an explanation to page 5 line 143 to 144:
Figure 5 shows the dependency of the resonant frequencies on the design parameter b (parameter a was fixed at 27 µm, distance between rings was always 50 µm as shown in Table 1, total length was 54 µm for that elliptic spoke need to connect with rings).
- The analytical evaluation of the equivalent stiffness, based on straight beams is very rough. Actually it is possible to properly evaluate the stiffness using beam theory of curved beams.
Thank you for your valuable comments. We agree that the beam theory of curved beams can explain the stiffness more precisely. However, in this research we mainly focused on the design and experimental evaluation. We think the beam theory can be the future research topic about analyzing the effective stiffness of the whole structure using elliptic spokes.
- Figure 8 is not very clear from a graphical point of view: the point of frequency matching cannot be detected.
To enhance the readability of the figure, we would like to modify the figure as shown in the modified manuscript.
- There are a lot (10 out of 39!!) self-citations. Some of them are only loosely related to the contents of the paper. I strongly suggest to reduce the number of self-citations.
We modified the references and make them more suitable for this manuscript. Please check in the modified manuscript.
Reviewer 3 Report
In the paper the authors proposed a new multi-ring disk resonator design with elliptical spokes. The use of elliptical spokes makes it possible to compensate for the elastic anisotropy (100) of single-crystal silicon and to use this material for the production of MEMS. The paper describes the manufacturing technology of such a resonator. Simulation and experimental results are presented, which confirm the effectiveness of the proposed approach. The research was done at a high level, it can be useful for specialists working in the field of MEMS. The paper may be published in the Special Issue Advanced Sensors in MEMS.
There are a few small remarks.
1. In the paper it is useful to present the requirements for the allowable value of the frequency mismatch in which the proposed resonator can be used in MEMS.
2. It is useful to add in Table 1 the physical characteristics of single-crystal silicon that were used in the simulation.
3. It is useful to add a comment to Figure 6 in section 2.2.2 and explain why QTED increases with the growth of design parameter b.
4. Figure 12 shows the design of the experimental setup. It is not clear from this figure how the resonator was fixed during the experiments. This can be further shown at the bottom of Figure 12.
Taking into account these small remarks will improve the paper.
Author Response
First, we would like to thank you for facilitating the timely and high quality review process. Each of our responses to the comments from you as well as modifications to the manuscript is attached below.
- In the paper it is useful to present the requirements for the allowable value of the frequency mismatch in which the proposed resonator can be used in MEMS.
Thank you for your nice comment on the allowable value of the frequency mismatch. We would like to add explanation in page 14 line 307-312.
The small remaining frequency mismatch less than 2000 ppm and Q-factor mismatch could be compensated by electrostatic tuning [35, 36] or feedback control [7]. The reduced frequency mismatch as small as 0 could be reported [26], which can completely meet the requirement for FM gyroscopes. The proposed resonator can be also applicable for a rate integrating gyroscope [37-39], which also requires the degenerated resonator.
- It is useful to add in Table 1 the physical characteristics of single-crystal silicon that were used in the simulation.
We would like to add the density, thermal conductivity, number of rings in the manuscript according to your comments.
|
Proposed resonator (with elliptic spokes) |
Conventional resonator (with linear beams) |
Thickness |
50 µm |
|
Density |
2330 kg/m3 |
|
Thermal Conductivity |
130 W/m/K |
|
Number of Rings |
10 |
|
Width of Rings |
10 µm |
|
Distance between Rings |
50 µm |
|
Radius of Centre Structure |
500 µm |
|
Resonant Frequency Mismatch |
70 ppm |
44000 ppm |
- It is useful to add a comment to Figure 6 in section 2.2.2 and explain why QTED increases with the growth of design parameter b.
We would like to add an explanation in page 6 line 164-166:
Therefore, frequency reduction caused by the design parameter b (a is fixed), increased QTED. For that in isothermal region, 1/QTED is proportional to the frequency [32].
- Figure 12 shows the design of the experimental setup. It is not clear from this figure how the resonator was fixed during the experiments. This can be further shown at the bottom of Figure 12.
Thank you for your nice comment on the experimental setup schematic. We would like to modify the figure as shown in the modified manuscript.